# Clinical Characteristics of Suicidal Youths and Adults: A One-Year Retrospective Study

**DOI:** 10.3390/ijerph17238733

**Published:** 2020-11-24

**Authors:** Vincent Besch, Christian Greiner, Charline Magnin, Mélanie De Néris, Julia Ambrosetti, Nader Perroud, Emmanuel Poulet, Martin Debbané, Paco Prada

**Affiliations:** 1Clinical Developmental Psychology Unit, Faculty of Psychology and Educational Sciences, University of Geneva, 1205 Geneva, Switzerland; Martin.Debbane@unige.ch; 2Unit for Investigation and Brief Treatment, Geneva University Hospitals, 1205 Geneva, Switzerland; christian.greiner@hcuge.ch (C.G.); melanie.deneris@hcuge.ch (M.D.N.); julia.ambrosetti@hcuge.ch (J.A.); paco.prada@hcuge.ch (P.P.); 3Emergency Psychiatry Department, Edouard Herriot Hospital, 69003 Lyon, France; charline.magnin@chu-lyon.fr (C.M.); emmanuel.poulet@chu-lyon.fr (E.P.); 4Department of Psychiatric Specialties for Emotional Regulation Disorders, Geneva University Hospitals, 1201 Geneva, Switzerland; nader.perroud@hcuge.ch

**Keywords:** young, youth, suicide, suicidality, suicidal youth, borderline personality disorder, psychiatric emergency, crisis

## Abstract

Suicide is a major mental health problem, particularly during youth, when it is the second leading cause of death. Since young people at risk of suicide are often cared for by the adult health system, we sought to identify the specificities and similarities between suicidal youths and adults in order to further inform the potential need for adaptations in taking care of suicidal youths. For this study, we used the following data: mental disorders, treatments, previous hospitalization, and reasons for current hospitalization, that were collected from November 2016 to October 2017 among people hospitalized for a suicidal crisis in a specialized psychiatric unit. First, we compared the data from the youth group with those from the adult group, and then we tried to determine if there were any associations between variables. Analyses showed that youths were more similar to adults than expected. In particular, we found comparable rates of personality disorders (especially borderline) and relapse, and similar profiles of reasons for hospitalization in suicidal crisis. Remarkably, among youth, neuroleptics appeared to be associated with fewer hospitalizations for behavioral than ideational reasons, but with more relapses. Results of this study suggest that young people could benefit from brief psychotherapeutic interventions implemented for adults.

## 1. Introduction


*“This is the end … and all the children are insane”.*


In 1967, Jim Morrison with his incantatory voice sang these weird, disturbing lyrics, expressing his perception of a termination process in a famous song of his rock band *The Doors*. Was he talking about his own end, or the end of a community whose children were developing mental disorders? It was not said. However, four years later, at the age of 27, he took his own life. Fifty years after, youth is considered as the critical period of life for mental health [1], and the rate of mental disorders in adolescents dying from suicide was estimated up to 98% [2]. Jim Morrison’s lyrical association between suicide and mental health in youth appears to be supported by scientific evidence. Clinical and scientific interest in this association has been stimulated by the observation that, while research on the other major causes of premature death and on psychopathology in childhood and adulthood is common, studies linking mental health and suicide during the pivotal period of youth are less frequent [3].

Suicide in young people is a critical issue for several reasons. It is the second leading cause of death among youth after road injury [4], and its prevalence has increased by 30% from 2000 to 2016 in the US [3]. Suicidality is frequently associated with mental disorders, such as mood disorders, schizophrenia, substance use disorder, anxiety, personality disorders [5,6,7], as well as with adverse social and interpersonal conditions [5,6,8,9], making it a phenomenon that is difficult to predict and to prevent [10]. Suicide attempts should be seen as the visible consequence of a psychological dynamic rather than like a single self-contained behavior: the suicidal dynamic is sustained by interacting psychological, life history, and social conditions and risk factors [3,11,12]. Furthermore, recent findings suggest that protective factors such as emotional intelligence and regulation skills also contribute to the suicidal dynamic [13,14,15].

Recent research has proposed to consider suicide attempts as the result of a three-phase process [16]. The first phase entails biological, environmental, and life-events as background factors (pre-motivational phase). During the second phase, a combination of feelings drives the subject to form suicidal ideations (ideation formation). At the third phase, volitional moderators enable the transition from ideas to action and to suicide attempt (behavioral phase). According to personal factors, such as impulsivity or acquired ability to overcome fear, the second phase can be bypassed to reach the behavioral phase directly. When a suicide attempt fortunately fails, it can have lasting consequences on health [17], particularly suicidal relapse, which means that a previous suicide attempt is the first risk factor of subsequent death by suicide. From an empirically informed lifespan perspective, Slama et al. [5] propose that the age of first suicide attempts is accurately modelled by a mixture of two Gaussian distributions (a Gaussian distribution, also called normal distribution or “bell curve”, is a probability distribution law, centered at its mean or average, and characterized by its variability, or standard deviation): the first centered at age of 19.5 years with a standard deviation (SD) of 4.3 years, and the second at age of 38.5 years with SD of 12.4 years. Remarkably, the probability density function of the first peak is twice as high as that of the second peak (0.035 vs. 0.018), and the age limit between the two subgroups of first-time suicide attempts is 26 years.

In the perspective of McGorry [18], this study considers youth as ranging from the end of childhood until mid-twenties, i.e., roughly from 12 to 25. This period overlaps adolescence and young adulthood and is crucial for mental health. As youth encompasses both initial suicide attempts and initial onset of mental disorders, it appears warranted to frame suicidality within a developmental model of psychopathology. Indeed, 75% of mental disorders appear before the age of 24 [1], and most syndromes emerge during the transition from puberty to mid-twenties [18,19]. Taking into account the symptoms and treating the disorders of youths with appropriate care may significantly reduce the detrimental evolution of mental and somatic health, may prevent the impact on key areas such as education, interpersonal relations, social and professional integration, and may divert them from developmental trajectories that would increase the likelihood of premature death [18]. However, young people are exposed to a break in the continuity of care because health services are often organized distinctively for children/adolescents and for adults, with a threshold around the age of 18 [20], and when in suicidal crisis, young people can be admitted to adult services.

This study takes place in a broader quality of care project of a psychiatric emergency department taking in care patients aged 16 and above, and which goal is to improve clinical practice through a better comprehension of their needs. It focuses on young patients, with aims to describe their clinical characteristics in comparison with those of adults, and to identify their clinical needs in order to inform potential adaptations to personalized interventions. The questions that we specifically address in this study are: (1) what disorders do they present at admission in psychiatric emergency? (2) Which treatments were they taking prior to admission? (3) Have they already been hospitalized in a psychiatric unit? (4) What are the reasons that justify the present hospitalization?

This study is theoretically and empirically framed on Sroufe and Rutter’s principles of developmental psychopathology [21], on the results of Kessler [1] and Jones [22] regarding mental disorders’ age of onset and development through life course, as well as on McGorry’s contemporary perspective of needs for youth mental health [18]. Sroufe and Rutter’s principles of developmental psychopathology [21] are especially relevant when considering age-related differences in mental disorders. For instance, results from life course studies indicate that anxiety and conduct disorders usually appear earlier in childhood [22,23], whilst mood, substance use, and personality disorders appear later during adolescence and further develop between the 20s and 40s [1,24,25]. Across the lifespan, suicidal behavior may represent a maladaptive coping strategy more likely activated in stressful situations. This illustrates both homotypic and heterotypic continuity in a developmental psychopathology framework.

As regards personality disorders (PDs), although studies indicate that they can be reliably assessed in young people [25,26] and that they represent an important risk factor for suicidality [27,28], most studies examining the repartition of mental disorders with age do not include diagnoses of PD [1,19,29,30]. Indeed, research documents the reluctance of clinicians to give youths PD diagnoses [23,31,32]. From the perspective of developmental studies that do report on PDs, it appears that their prevalence increases from youth to adulthood [24].

Based on epidemiological and developmental psychology studies, this study’s expectations are that PDs should be less frequently diagnosed in youths in comparison to adults presenting at emergency psychiatric services. Furthermore, it is expected that anxiety disorder should be more frequent in youth, whereas substance use disorders less frequent. Finally, it is expected that frequency of mood disorders should be equivalent in youths and adults presenting at the psychiatric emergency service [29,30,33].

In terms of treatment, it is expected that youths would present at psychiatric emergency in similar ratios of ongoing psychotherapeutic treatment in comparison to adults [34], and generally, with less ongoing pharmacological treatments [35,36], in particular less neuroleptics [37,38] and less antidepressants due to warnings on their iatrogenic effect in suicidal youths [39]. Concerning previous hospitalizations, despite the logical fallacy that *the older you are the more time you had to relapse*, research indicates that suicidal relapse is as frequent among young people as it is among adults [40,41]. Finally, concerning reasons of hospitalization, higher prevalence of impulsivity and conduct disorders in youths support the hypothesis that they would present at the psychiatric emergency more frequently for suicidal behaviors rather than suicidal ideations, while adults would have a more balanced distribution between behaviors and ideations [42].

Adoption of an age-comparative approach has been done in previous studies but, to our knowledge, the set of variables we have chosen and the search for associations between them is unprecedented.

## 2. Materials and Methods

### 2.1. Study Population

Our sample consisted in the population who was admitted at Geneva University Hospital (HUG) and taken into care by the team of the Unit for Investigation and Brief Treatment (UITB). HUG covers the great Geneva urban area, with 500,000 inhabitants in 2018. It captures all psychiatric emergencies. In a triage mode, patients who are at risk of suicide are oriented to the UITB, a psychiatric emergency unit dedicated to the management of patients in suicidal crisis. Exceptions are for people with somatic or psychiatric diseases that cannot be supported in this unit, namely: psychotic decompensation, manic or hypomanic state, substance use/addiction and withdrawal demand, and severe psychomotor agitation. The UITB welcomes between 350 and 400 people each year, with the mission to investigate the disorders underlying suicidal crisis and to initiate remedial treatments in a brief stay with an average length of 6 days. Patients must be 18 years of age or older to be admitted, but patients aged 16 and 17 are admitted as long as it is possible to provide them with a single room. In practice, two thirds of the patients admitted to UITB are addressed by health professionals or other HUG internal or external services, while one third are brought after intervention of emergency medical service, or by relatives, or present themselves spontaneously.

In our study, we have considered all patients admitted, without restriction, for a total of 357 patients aged 16 to 74 years. In a cross-sectional approach and following literature on the critical period for the onset of mental disorders, and on the age of first suicide attempts, we have considered two age groups: patients aged 16 to 26 years, whom we called “youths”, and patients aged 27 years and above, whom we called “adults”.

### 2.2. Data Collection and Management

Retrospective data from 1 November 2016 to 31 October 2017 was collected for each patient hospitalized in the UITB for suicidal crisis: the principal and secondary psychiatric diagnoses, the existence and type of ongoing pharmacological or psychotherapeutic treatments, the existence of previous psychiatric hospitalization, and the reasons that justified the current hospitalization. Data were anonymized when transmitted for research purposes.

Based on the collected raw data, we have considered the following four variables of interest:

Mental disorders: based on the main and secondary diagnoses evaluated by UITB psychiatrists. After controlling data, we have grouped related diagnoses into 9 mental disorders: anxiety, depressive disorder, bipolar disorder, borderline personality disorder (BPD), personality disorder other than borderline (OPD), psychotic spectrum disorder (PSD), post-traumatic stress disorder (PTSD), eating disorder, substance use disorder (SUD).

Ongoing treatments: at admission, patients were asked about the treatments they were undergoing considering three types of medicated treatments: antidepressants, anxiolytics, and neuroleptics, as well as psychotherapeutic treatments.

Previous psychiatric hospitalization (PPH): at admission, patients were asked if they had already been hospitalized in a psychiatric unit. We were also informed by UITB administrative records of patients with previous stay in this unit, i.e., with a previous psychiatric hospitalization for unambiguous suicidal crisis.

Reasons that caused the current hospitalization: formulated by the psychiatrist who took decision for admission. They were based on patient behaviors reported by her/him or those who brought the patient to the emergency room, as well as on the ideas and intentions s/he expressed, and on the affective and cognitive states s/he exhibited. These reasons were intended to capture the motives that led to decide on hospitalization for suicidal crisis, and to express them in common terms. As Table 1 shows, reasons for hospitalization collected from hospital records were numerous. In order to operationalize these into a categorical variable useable for statistical analysis and consistent within a theoretical framework [16], we classified them into 3 categories: (1) the presence of suicidal or self-destructive behaviors, (2) the verbal expression of suicidal thoughts or intentions, and (3) the presence of symptoms severe enough to justify hospitalization for risk of suicide but without explicit behaviors nor ideations. On this basis, we defined a nominal variable ‘reason’, with 3 modalities: Behaviors (SB), Ideations (SI), and Other acute signs (OAS) when there were no behaviors nor ideations, but the acute presence of severe symptoms that warn for the risk of acting out (see Table 1). In practice, admission reports frequently indicated more than one reason, for examples ‘depressive mood and suicidal ideas’ or ‘anxiety and depression’. Following the phased model of suicidality mentioned above [16], for a patient, we retained the reason that was the closest to suicide attempt, going from risk factors (mainly affective or dissociative symptoms) considered as other acute signs, to explicit ideations, and finally self-destructive behaviors. In the examples above, we retained respectively “Ideations” and “Other acute signs”.

Rather than considering some of these variables as causes and others as consequences, we chose to consider them as contributing simultaneously and possibly interacting in the complex process that underlies suicidality. In other words, we regarded suicidal crisis as a possible result of interactions between these variables. Therefore, in a first step, we analyzed and described our variables of interests for both age groups and looked for common features and differences between groups. In a second step, we looked for associations between variables of interest in the two age groups and compared these associations between groups. Figure 1 shows how we have organized and analyzed our variables and their relationships.

### 2.3. Statistical Analysis

Our objective was to characterize profiles of suicidal patients according our variables of interest (Figure 1), and to look for associations between those variables, with consideration of 2 age groups: the youths aged 16 to 26, and the adults aged 27 and above.

Aligned with our variables of interests, we defined 9 binary variables for mental disorders, set at 1 or 0 according to disorders presence or absence. For treatments, we defined 4 binary variables, set at 1 or 0 according to patient exposure or not to treatments. Previous psychiatric hospitalization was directly captured from admission interview (binary at 1 or 0). The variable ‘reason’ was nominal with 3 modalities: other acute signs, ideations, behaviors (Table 1).

In a first step, we aimed to describe our two groups through calculation of frequencies for all variables within groups, with test of equifrequency when relevant, and test of independence between groups. We used Pearson Khi2 for test independence or adjustment, and calculated odds ratio (OR) and confidence interval at 95% (95% CI). Please note that in the following, the degree of freedom (df) is 1 except where specifically indicated.

In a second step, we evaluated the existence of associations between variables within age groups. For this, we calculated frequencies for 6 combinations of 2 variables (represented by the arrows in Figure 1) and calculated the odds ratio at 95% confidence interval. We graphically summarized the meaningful associations for each group to facilitate comparative analysis between groups.

All calculations were made using Excel Microsoft Office Professional Plus 2016, using statistic and logarithmic functions.

## 3. Results

Prior to our planned analysis, we examined the number of patients admitted for suicidal crisis according to age, and we inspected the relation between previous psychiatric hospitalization and previous stay at UITB, i.e., of unambiguous previous suicidal crisis (Figure 2). The curve of the average number of hospitalized patients per age range (grey) shows a pattern consistent with a previous study [5], with a peak around 20 years old, then a plateau, before a decrease around 36 years old. The same curves only for patients with previous psychiatric hospitalization (orange) or with previous stay at UITB (blue) show a strong parallelism, suggesting that those two conditions are closely related. Indeed, statistical tests indicated a strong association between previous psychiatric hospitalization and previous stay at UITB: Khi2 = 217.6, *p* < 0.001, and OR = 305.7, 95% CI 71.9–1299.2.

We also checked gender distribution and examined comorbidity. Women were more represented than men in both groups: 77.1% for youth (Khi2 = 30.9, *p* < 0.001) and 68.7% for adults (Khi2 = 35.1, *p* < 0.001), and sex ratios were similar between the groups (Khi2 = 2.60, *p* = 0.107). Considering comorbidity for patients with more than one disorder, we found no difference between age groups: among youth, 39% had more than one disorder and 48.4% of adults (Khi2 = 2.62, *p* = 0.106). We observed an interaction between age and comorbidity with regards to previous psychiatric hospitalization: while youth with previous psychiatric hospitalization were 37.5% more likely to have more than one disorder, this rate rose to 63% for adults (Khi2 = 6.30, *p* = 0.01, OR = 2.84, 95% CI 1.24–6.53).

### 3.1. Observed Frequencies of Variables of Interest per Age Group

Table 2 summarizes frequencies of disorders, treatments, previous psychiatric hospitalization, and reasons for hospitalization observed in our population. The dominant disorder observed in both groups was depression (Figure 3), which was more frequent in adults (67.1%) than in youth (51.4%, Khi2 = 7.73, *p* = 0.005). It was followed by personality disorders (18.1% in youth, 21% in adults), especially borderline (17.1% and 13.5%) with no difference between age groups (respectively Khi2 = 0.397, *p* = 0.529, and Khi2 = 0.794, *p* = 0.373). The third most frequent disorder was Post-Traumatic Stress Disorder (PTSD), higher in the youth group (29.5%) than in the adult one (16.3%, Khi2 = 8.09, *p* = 0.004). Substance use disorder was tendentially higher for adults (13.1%) than for youths (6.7%, Khi2 = 3.08, *p* = 0.079).

The most frequent treatment reported was anxiolytics (Figure 4), then psychotherapy, and at last antidepressants and neuroleptics. Psychotherapy (youth 61.9% and adults 63.1%) and neuroleptics (32.4% and 30.2%) were equally observed in both groups (Khi2 = 0.017, *p* = 0.897, and Khi2 = 0.119, *p* = 0.73). Antidepressants were more frequent in adults (54.8%) than in youths (28.6%, Khi2 = 10.8, *p* = 0.001). Anxiolytics were tendentially more frequent in adults (82.1%) than in youths (64.8%, *Khi2 =* 2.91, *p* = 0.088). Within groups, frequencies of treatments were different (for youth, Khi*2 =* 24.4, df = 3, *p* < 0.001, for adults Khi*2 =* 61.0, df = 3, *p* < 0.001).

Frequency of previous psychiatric hospitalization was 30.5% for youths, and 36.5% for adults, with no difference between groups (Khi2 = 1.19, *p* = 0.275).

In both groups, the main reason for hospitalization (Figure 5) was ideations (50.5% in youth, 51.2% in adults), followed by behaviors (28.6% and 26.6%), and lastly other acute signs (21.0% and 22.2%). Within groups, frequencies per reason were different (for youths Khi*2 =* 14.8, df = 2, *p* < 0.001, for adults Khi2 *=* 36.9, df = 2, *p* < 0.001), but their distributions were similar between groups, i.e., reasons for hospitalization were independent of the age (Khi2 = 0.17, df = 2, *p* = 0.919).

### 3.2. Associations between Variables of Interest

#### 3.2.1. Associations between Disorders and Treatments

Table 3 gives the frequencies of disorders per treatment exposure and odds ratio within age group. For the youth group, patients with antidepressants showed more depression (70% vs. 40%, OR 3.50, 95% CI: 1.4–8.74, *p* = 0.007), those taking anxiolytics less Other Personality Disorders than Borderline (OPD, 11.8% vs. 34.4%, OR 0.255, 95% CI: 0.09–0.72, *p* = 0.01), while neuroleptics were more frequently associated with BPD (35.3% vs. 8.8%, OR 5.64, 95% CI: 1.89–16.8, *p* = 0.002), and psychotherapy with OPD (24.6% vs. 7.7%, OR = 3.92, 95% CI: 1.06–14.5, *p* = 0.04) and tendentially with BPD (23.1% vs. 7.7%, OR 3.6, 95% CI: 0.97–13.4, *p* = 0.056).

For the group of adults, patients under antidepressants showed more depression (81.2% vs. 50.0%, OR 4.31, 95% CI: 2.43–7.63, *p* < 0.001) and less PTSD (8.7% vs. 26.4%, OR 0.265, 95% CI: 0.13–0.55, *p* < 0.001), PTSD was less frequent for patients taking anxiolytics (13% vs. 33.3%, OR 0.30, 95% CI: 0.14–0.65, *p* = 0.002), neuroleptics was associated with more frequent Bipolar disorder and BPD (11.8% vs. 1.8%, OR 7.39, 95% CI: 1.94–28.1, *p* = 0.003 and 22.4% vs. 9.5%, OR 2.74, 95% CI: 1.3–5.77, *p* = 0.008), and psychotherapy was associated with more BPD (18.9% vs. 4.4%, OR 5.0, 95% CI: 1.7–14.7, *p* = 0.003) and less substance use disorder (9.4% vs. 20.0%, OR 0.417, 95% CI: 0.20–0.87, *p* = 0.021).

#### 3.2.2. Associations between Disorders and Previous Psychiatric Hospitalization

Table 4 shows that in the group of youths, previous psychiatric hospitalization was associated only with BPD (OR = 3.69, 95% CI: 1.3–10.5, *p* = 0.015). In the group of adults, previous psychiatric hospitalization was associated with BPD (OR = 6.26, 95% CI: 2.77–14.1, *p* < 0.001), negatively with PTSD (OR = 0.31, 95% CI: 0.13–0.72, *p* = 0.007), and tendentially with substance use disorder (OR = 2.04, 95% CI: 0.98–4.27, *p* = 0.058).

#### 3.2.3. Associations between Disorders and Reasons for Hospitalization

Due to small subgroups size, we could only assess associations of reasons for hospitalization with depression and PTSD for youth, as well as with BPD, OPD, and substance user disorders for adults. The results shown in Table 5 suggest no association between disorders and reasons for hospitalization in both age groups. This is consistent with the independence of the reasons for hospitalization in suicidal crisis and mental disorders.

#### 3.2.4. Association between Treatments and Previous Psychiatric Hospitalization

Table 6 shows that in our group of youths, previous psychiatric hospitalization was associated with more anxiolytics (OR = 3.14, 95% CI: 1.07–9.19, *p* = 0.037), neuroleptics (OR = 3.86, 95% CI: 1.58–9.42, *p* = 0.003), and psychotherapy (OR = 4.53, 95% CI: 1.57–13.1, *p* = 0.005). In the group of adults, previous psychiatric hospitalization was associated with neuroleptics (OR = 1.95, 95% CI: 1.12–3.40, *p* = 0.018) and with psychotherapy (OR = 3.48, 95% CI: 1.9–6.37, *p* < 0.001).

#### 3.2.5. Association between Treatments and Reasons for Hospitalization

Tests of adjustment shown in Table 7 indicated associations between treatments and reasons for hospitalization in two conditions. First, while the average frequency of neuroleptics in the youth group (32.4%) is similar to what is observed in the group of adults (30.2%, Khi*2 =* 0.16, *p* = 0.69), there is a significantly lower rate of suicidal behaviors in youth with neuroleptics: only 13.3% were hospitalized for suicidal behaviors against 42% for ideations and 40.9% due to warning signs (Khi*2 =* 7.66, df = 2, *p* = 0.02). Second, antidepressants were taken by 49.2% of adult patients with ideations, 60.9% of those with behaviors, and 68.5% of those hospitalized due to suicidal other acute signs (Khi*2 =* 6.42, df = 2, *p* = 0.04).

In the youth group, complementary odds ratio calculation indicated that neuroleptics were associated with both more ideations and other acute signs than behaviors (OR = 4.7, 95% CI 1.73–12.8, *p* = 0.011 and OR = 4.5, 95% CI 1.45–14.0, *p* = 0.029), and conversely less behaviors than ideations and other acute signs (OR = 0.21, 95% CI 0.08–0.57, *p* = 0.009). This is consistent with the existence of an association between neuroleptics and reason for hospitalization for youth in suicidal crisis in the population of our study. The use of neuroleptics is associated with lower rate of hospitalization motivated by suicidal behaviors.

In the group of adults with antidepressants, odds ratio indicated no difference between other acute signs and behaviors (OR = 0.72, 95% CI 0.38–1.36, *p* = 0.39), but a difference between other acute signs and ideations (OR = 0.44, 95% CI 0.25–0.78, *p* = 0.018). This is partially consistent with an association between antidepressants and less verbalized symptoms of suicidal risks.

#### 3.2.6. Association between Reasons for Hospitalization and Previous Psychiatric Hospitalization

Table 8 gives the frequencies of reasons for hospitalization for patients with previous psychiatric hospitalization. In both age groups, among people with previous psychiatric hospitalization, reasons for hospitalization were unevenly distributed (Figure 6a,b): suicidal ideations came first (50.0% for youth, 47.8% for adults), suicidal behaviors second (respectively 37.5% and 31.5%), and other acute signs third (12.5% and 20.7%). Reasons for current hospitalization did not vary according to age nor to previous hospitalization.

Figure 7a (youth) and b (adults) illustrates our findings regarding associations between variables for both groups. Associations are not directional, arrows in the legends indicate in which direction the odds ratios play, e.g., in the youth group, there is an association between disorder and previous psychiatric hospitalization, namely, BPD was found associated with an OR of 3.7 with previous psychiatric hospitalization, i.e., we observed in our study that youth with BPD had 3.7 more chance to have had previous psychiatric hospitalization.

## 4. Discussion

In this study, we investigated the characteristics of suicidal patients in terms of mental disorders, pharmacological and psychotherapeutic treatments, previous psychiatric hospitalization, and reasons leading to their hospitalization. Our goal was to compare the profile of young people aged 16 to 26 years to adults aged 27 and more. We also examined associations between variables to contribute to the understanding of suicidality in youth and investigate their needs for care in comparison with adults.

Across our cohort of 357 patients, we observed a higher prevalence of hospitalized women vs. men (77.1% in youth, 68.7% in adults), making an average female to male ratio of 2.4, which is the inverse of the death by suicide reported ratio of 2.2 men for 1 woman observed in Switzerland [35]. This “gender paradox” has already been described in studies on adults and youths [43,44], which propose that factors out of the scope of this study are involved: socio-economical context and life events for women, and access to means and impulsivity for men.

In accordance with our hypotheses, depression was the most frequent disorder in youths and adults, and substance use disorder was tendentially less present among youths. With regards to treatments, antidepressants were less frequent in youths, while psychotherapy was equally present in both groups. Rates of previous psychiatric hospitalization were similar in both groups. Of note, PTSD was frequent in both groups, remarkably in youths with almost 30% prevalence, while it is seldom or only indirectly mentioned in other studies, and at much lower rates [16,27]. This suggests personal histories of trauma, which could be related to adverse childhood conditions or traumatic life-events.

Opposite to expectations, depression prevalence was lower in youths than in adults, whereas personality disorders were similar in both groups. With regards to prescription of psychopharmacological treatments before hospitalization, neuroleptics prescription in youths appeared equivalent as in adults, while it was expected to be lower. Regarding the distribution of the reasons for hospitalization (suicidal behaviors versus suicidal ideas), these were comparable between youths and adults, while youths were expected to exhibit more behaviors than ideas. While not confirming predictions, these findings may be consistent with research indicating that suicidality is transdiagnostic in both youths and adults: in our population, it appears related with different mental disorders, which are addressed with different types of pharmacological or psychotherapeutic treatments. The average number of hospitalizations per age range observed in our population (Figure 2) is consistent with an onset of suicidality during youth. A rate of rehospitalization around 30% is consistent with a substantial chronicity. This chronicity is especially marked for patients with a diagnosis of borderline personality disorder [45], who have, respectively among youths and adults, a 3 and 6 odds ratio of rehospitalization in comparison with those without this diagnosis. Taken together, our observations highlight that suicidal youths exhibit clear clinical needs. In particular, they exhibit equally frequent personality disorders (especially borderline), receive neuroleptics prescriptions just as frequently, and experience equivalent rehospitalizations. Furthermore, the results yield clinical specificities of youths: higher frequency of PTSD, less diagnosed depression, and less prescribed antidepressants, which argue for further research into this population consulting in emergency psychiatry settings.

The higher prevalence of BPD diagnosis in youths suggests that the clinicians in the current study were more inclined to use this diagnosis with youths in comparison to practices reported elsewhere [31]. The higher rate of neuroleptics may result from a shift in the prescription of antidepressants to neuroleptics in an attempt to reduce the severity of presenting symptoms [46,47]. Findings from the present study suggest that neuroleptics prescription is associated in youths to less frequent hospitalization due to suicidal behavior in comparison with ideations and other acute signs, while this is not the case in adults. Yet, it is not clear that neuroleptics may have reduced suicidal behavior in itself, and this hypothesis would need specific research designs to be tested.

When considering associations between rehospitalization and treatments, one can note that rehospitalization is associated with prescription of neuroleptics and psychotherapy. This observation questions the efficiency of these treatments on the underlying distal causes that drive the suicidal process. Further research should help to better understand the effects of such treatments on suicidal behavior vs. ideations. In order to bring some understanding on the complex suicidal mechanisms, future research could make distinctions between the types of neuroleptics, and between psychotherapeutic approaches with different targets.

Interestingly, we found that 70% of young people taking antidepressants are diagnosed with depression, while only 38% of those with neuroleptics receive this diagnosis. Following Kessing’s results [48] showing that suicidal relapse is correlated with the severity of the depressive symptoms, and suggesting an association with personality disorders, it is conceivable that in the presence of borderline personality disorder, an episode of major depression may foster suicidal intent. This hypothesis should attract further detailed scrutiny in future research.

Linking back to clinical practice, this hypothesis would bolster the importance of an accurate evaluation, without the fear of diagnosing personality disorders (which looks already acquired), and with clear identification and distinction of depressive symptoms, as depression remains the disorder most associated with suicidal crisis (Table 2) and its relapse (Table 4). In terms of treatment, given the difficulties to treat suicidality in youths with pharmacological means [38], and the fact that emotional dysregulation is an etiological factor in both depression and BPD [15,41,49], the implementation of psychotherapeutic interventions specifically targeting emotional regulation skills, with evidence-based results in adults [50], should be considered in young people.

## 5. Conclusions

### 5.1. Conclusions

In terms of mental disorders, previous psychiatric hospitalization, and reasons for hospitalization, the profiles of youth and adults of this study looked more similar than dissimilar. Nevertheless, some differences in medication have raised questions. Results show that the practice of diagnosing personality disorders in young patients is performed in the adult psychiatric emergency department examined in this study. The diagnosis of borderline personality disorder appears to be specifically associated with previous hospitalization. Concerning previous treatments, results suggest the need to better understand the mechanisms of pharmacological treatments in suicidal youth, given their association with a pattern of multiple hospitalizations. We also notice that, since depression and BPD seem to be entangled in suicidal populations rehospitalized in emergency, it might be interesting to further examine their common underlying etiological psychological processes. Indeed, evidence-based results of specialized psychotherapies indicate that different types of therapies target common, psychological, transdiagnostic processes such as emotion regulation, leading to significant reduction in suicidality [50].

### 5.2. Limitations

A limitation of our study is related to its exploratory approach: in some analyses on associations of variables, we sometimes arrived at subgroups too small for statistical significance, which is difficult to resolve other than with a larger cohort. The granularity of our data was sufficient to bring together the concepts of reasons for hospitalization, treatments, and rehospitalization, which made it possible to highlight the association, among young people, of the use of neuroleptics with a decrease in suicidal behaviors and an increase in rehospitalization. Nevertheless, one might have expected to see more associations between mental disorders and reasons, and one would be very interested in finding out what rehospitalization is associated with, for the usefulness this may have in the prevention of relapse. Therefore, it would be appropriate to deal with finer data in terms of psychiatric evaluations, i.e., dimensional symptoms rather than binary diagnoses, as well as of treatments. This would also enable to better assess associations between variables, e.g., with distinction of antidepressants types, or between first and second generations of neuroleptics, and to include mood stabilizers. Another limitation lies in the investigation of comorbidity which appeared significant but that we could hardly analyze with the statistical methods we have chosen in this study. Further factorial analysis might be interesting to perform, as a converging group of studies shows that this method reveals that a latent general factor of psychopathology might explain comorbidity as well as homotypic and heterotypic continuity of psychopathology. To further study suicidality, we could also work out a definition of ‘reasons of suicidal crisis’ that would take into account a wider scope of elements such as feelings, exposure, access to means, familial and social history and conditions, or the existence of a recent personal life event that may have triggered the crisis by exceeding patients’ capacities for affective and cognitive regulation and could explain the high frequency of PTSD we observed. Another limitation is the cross-sectional design of our study because it gave us no direct access to the relations between disorders, relapse, reasons for hospitalization, and treatments. If our findings suggest an association in youth between neuroleptics, relapse, and reasons, knowing what the reasons and the disorders at previous hospitalizations were would be necessary to describe the effect of treatments on changes in reasons and on evolutions in disorders.

Our findings are encouraging, and given the above limitations, replication studies with larger cohorts and multicentric design would be welcome to better understand the multifactorial process of suicidality, and to better care for young people in psychological and existential crisis.

## Figures and Tables

**Figure 1 ijerph-17-08733-f001:**
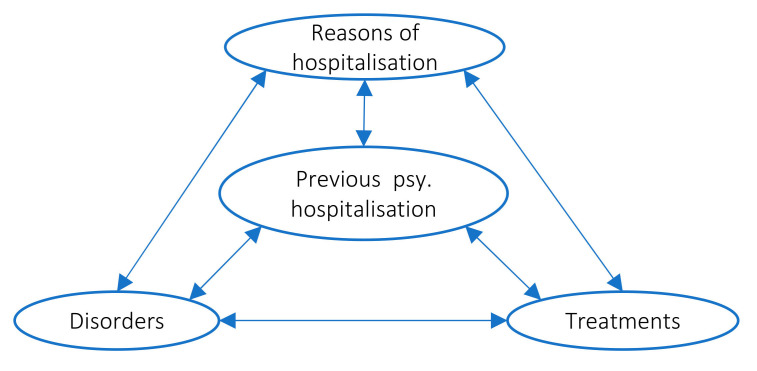
Organization of variables of interest and their associations.

**Figure 2 ijerph-17-08733-f002:**
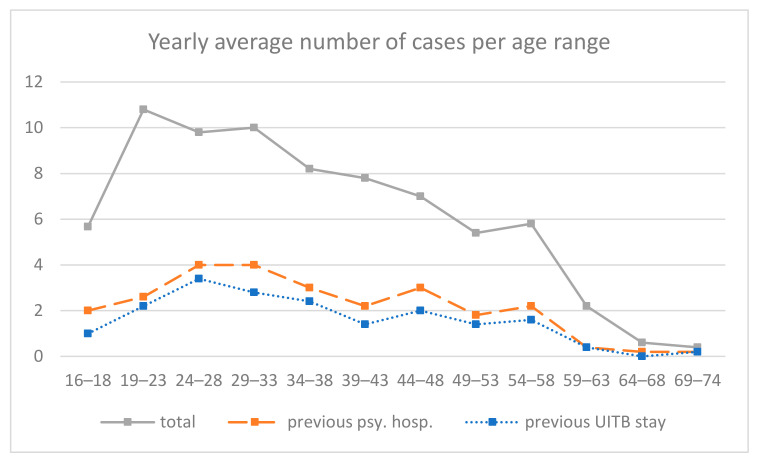
Distribution of suicidal crisis cases per age range from 16 to 74.

**Figure 3 ijerph-17-08733-f003:**
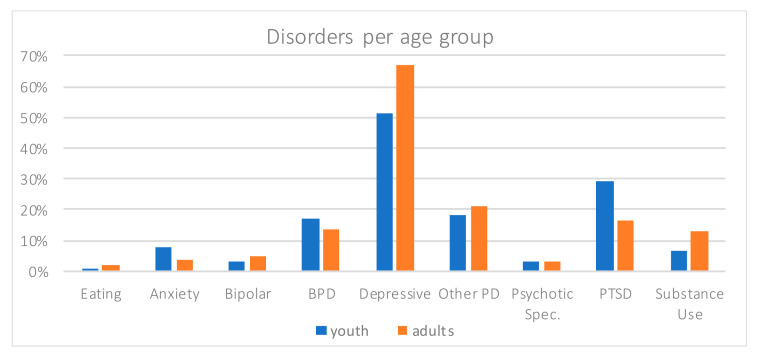
Disorders per age group.

**Figure 4 ijerph-17-08733-f004:**
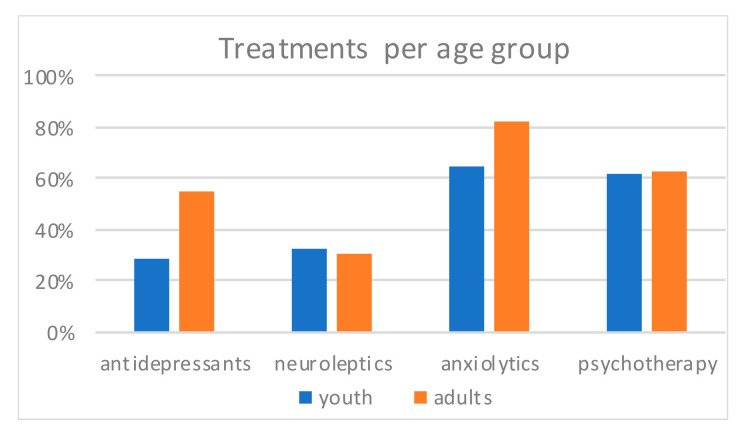
Treatments per age group.

**Figure 5 ijerph-17-08733-f005:**
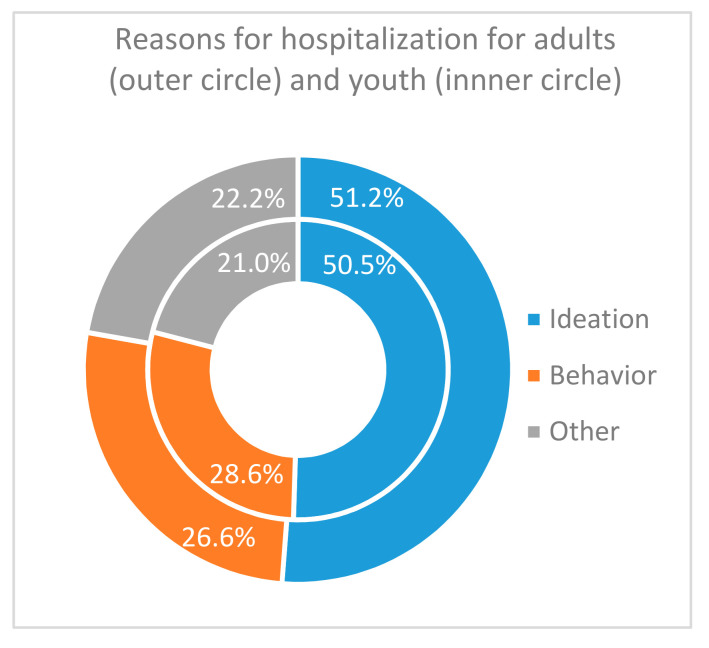
Reasons for hospitalization in suicidal crisis per age group.

**Figure 6 ijerph-17-08733-f006:**
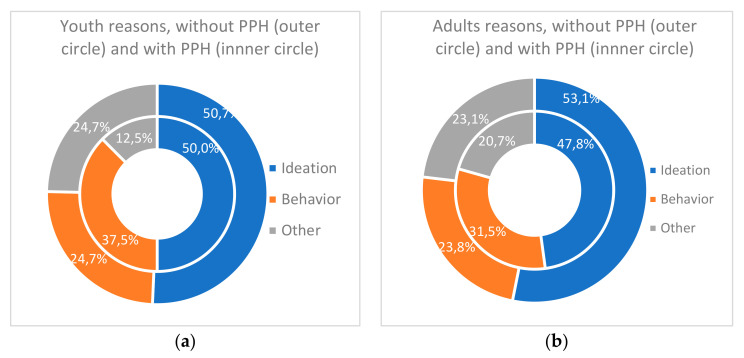
Frequencies of reasons for hospitalization according to previous psychiatric hospitalization (PPH), (**a**) for youth; (**b**) for adults.

**Figure 7 ijerph-17-08733-f007:**
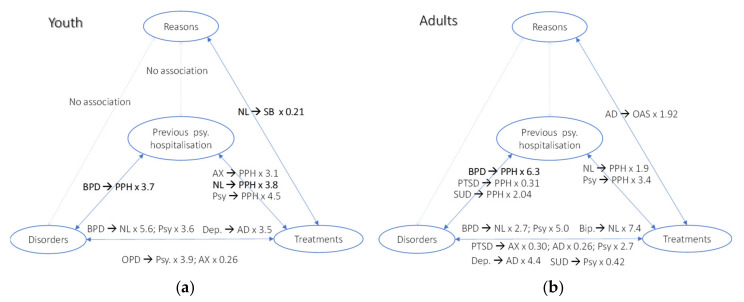
Relevant associations between variables of interest, especially in bold those with borderline personality disorder and neuroleptics. (**a**) in youth and (**b**) adults AD: Antidepressants, AX: Anxiolytics, Bip.: Bipolar Disorder, BPD: Borderline Personality Disorder, Dep.: Depression, NL: Neuroleptics, OAS: Other acute signs, OPD: Other Personality Disorders than BPD, Psy: Psychotherapy, PPH: Previous Psychiatric Hospitalization, PTSD: Post-Traumatic Stress Disorder, SB: Suicidal Behaviors, SUD: Substance Use Disorder.

**Table 1 ijerph-17-08733-t001:** Conversion of clinical reasons that motivated hospitalization into modalities of a standardized variable.

Clinical Reasons	Modality of Variable	Clinical Reasons	Modality of Variable
defenestration	Behavior	anxiety	Other acute signs
drowning	Behavior	auditory hallucination	Other acute signs
drug abuse	Behavior	bipolar disorder	Other acute signs
jump in front of a vehicle	Behavior	clastic crisis	Other acute signs
jump off bridge	Behavior	derealization	Other acute signs
substance overdose	Behavior	disorganization	Other acute signs
scarification	Behavior	dissociative stupor	Other acute signs
self-harm	Behavior	eating disorder	Other acute signs
self-injury	Behavior	exhaustion	Other acute signs
strangulation	Behavior	hallucination	Other acute signs
substance abuse	Behavior	inadapted speech	Other acute signs
suicide attempt	Behavior	manic	Other acute signs
wrist slashing	Behavior	depression	Other acute signs
dark thoughts	Ideation	mutism	Other acute signs
suicidal ideas	Ideation	panic crisis	Other acute signs
suicide ideation	Ideation	paranoid idea	Other acute signs
PTSD (Post-Traumatic Stress Disorder)	Other acute signs	persecution idea	Other acute signs
substance withdrawal	Other acute signs	psychotic crisis	Other acute signs
		psychotic decompensation	Other acute signs

**Table 2 ijerph-17-08733-t002:** Frequencies of variables of interests, test of independence, and odds ratio (OR) between age-group (-: no statistical significance).

Variables of Interest										
**Disorders**	**Youth**	***n***	**Adults**	***n***	**df**	**Khi2**	***p***	**OR**	**95% CI**	***p***
Eating	1.0%	1	2.0%	5	-	-	-	-	-	-
Anxiety	7.6%	8	4.0%	10	1	2.06	0.151	0.50	0.19–1.31	0.158
Bipolar	2.9%	3	4.8%	12	-	-	-	-	-	-
BPD	17.1%	18	13.5%	34	1	0.794	0.373	0.75	0.40–1.41	0.374
Depressive	51.4%	54	67.1%	169	1	7.73	0.005	1.92	1.21–3.06	0.006
OPD	18.1%	19	21.0%	53	1	0.397	0.529	1.21	0.67–2.16	0.529
PSD	2.9%	3	3.2%	8	-	-	-	-	-	-
PTSD	29.5%	31	16.3%	41	1	8.09	0.004	0.46	0.27–0.79	0.005
Substance Use	6.7%	7	13.1%	33	1	3.08	0.079	2.11	0.90–4.93	0.085
**Treatments Prior to Admission**	**Youth**	***n***	**Adults**	***n***	**df**	**Khi2**	***p***	**OR**	**95% CI**	***p***
Antidepressants	28.6%	30	54.8%	138	1	10.8	0.001	3.03	1.85–4.94	0.000
Neuroleptics	32.4%	34	30.2%	76	1	0.119	0.730	0.90	0.55–1.47	0.679
Anxiolytics	64.8%	68	82.1%	207	1	2.91	0.088	2.50	1.50–4.19	0.000
Psychotherapy	61.9%	65	63.1%	159	1	0.017	0.897	1.05	0.66–1.68	0.832
**Previous Psychiatric Hospitalization**	**Youth**	***n***	**Adults**	***n***	**df**	**Khi2**	***p***	**OR**	**95%CI**	***p***
	30.5%	32	36.5%	92	1	1.19	0.275	1.31	0.80–2.14	0.276
**Reasons for Hospitalization**	**Youth**	***n***	**Adults**	***n***	**df**	**Khi2**	***p***	**OR**	**95% CI**	***p***
Other acute signs	21.0%	22	22.2%	56	2	0.170	0.919	1.08	0.62–1.88	0.791
Ideations	50.5%	53	51.2%	129				1.03	0.65–1.62	0.902
Behaviors	28.6%	30	26.6%	67				0.91	0.55–1.50	0.701

BPD = Borderline Personality Disorder, OPD = Other than borderline Personality Disorder, PSD = Psychotic Spectrum Disorder, PTSD = Post-Traumatic Stress Disorder.

**Table 3 ijerph-17-08733-t003:** Frequencies of disorder per treatment exposure and odds ratio per age group (-: no statistical significance).

	Youth					Adults				
Antidepressants	No	Yes	OR	*p*	95% CI	No	Yes	OR	*p*	95% CI
**Disorder**										
Eating	1.4%	0.0%	-	-	-	1.9%	2.2%	1.16	0.87	1.90–7.04
Anxiety	4.3%	16.7%	4.47	0.051	0.99–20.8	3.8%	4.3%	1.16	0.82	0.32–4.22
Bipolar	2.9%	3.3%	-	-	-	4.7%	5.1%	1.08	0.9	0.33–3.50
BPD	17.1%	20.0%	1.21	0.73	0.41–3.59	14.2%	12.3%	0.85	0.67	0.40–1.80
Depressive	40.0%	70.0%	3.50	0.007	1.4–8.74	50.0%	81.2%	4.31	0.000	2.43–7.63
OPD	17.1%	23.3%	1.47	0.47	0.51–4.20	20.8%	21.7%	1.06	0.85	0.57–1.97
PSD	4.3%	0.0%	-	-	-	4.7%	2.2%	0.45	0.28	0.10–1.92
PTSD	35.7%	20.0%	0.45	0.125	0.16–1.25	26.4%	8.7%	0.26	0.000	0.13–0.55
Substance Use	8.6%	0.0%	-	-	-	15.1%	12.3%	0.79	0.53	0.38–1.65
**Anxiolytics**	**No**	**Yes**	**OR**	***p***	**95% CI**	**No**	**Yes**	**OR**	***p***	**95% CI**
**Disorder**										
Eating	0.0%	1.5%	-	-	-	0.0%	2.4%	-	-	-
Anxiety	9.4%	7.4%	0.77	0.73	0.17–3.43	0.0%	4.8%	-	-	-
Bipolar	6.3%	1.5%	-	-	-	5.1%	4.8%	0.94	0.94	0.20–4.46
BPD	12.5%	20.6%	1.81	0.33	0.55–6.03	10.3%	14.0%	1.43	0.53	0.47–4.31
Depressive	43.8%	51.5%	1.36	0.47	0.59–3.17	64.1%	68.6%	1.22	0.58	0.60–2.51
OPD	34.4%	11.8%	0.25	0.01	0.09–0.72	25.6%	20.8%	0.76	0.50	0.34–1.68
PSD	3.1%	2.9%	-	-	-	2.6%	3.4%	1.33	0.79	0.16–11.1
PTSD	34.4%	29.4%	0.79	0.62	0.32–1.95	33.3%	13.0%	0.30	0.002	0.14–0.65
Substance Use	3.1%	7.4%	2.46	0.42	0.28–22.0	10.3%	14.0%	1.43	0.53	0.47–4.31
**Neuroleptics**	**No**	**Yes**	**OR**	***p***	**95% CI**	**No**	**Yes**	**OR**	***p***	**95% CI**
**Disorder**										
Eating	1.5%	0.0%	-	-	-	0.6%	5.3%	9.28	0.05	1.02–84.5
Anxiety	5.9%	11.8%	2.13	0.31	0.50–9.11	4.2%	3.9%	0.94	0.94	0.24–3.76
Bipolar	0.0%	8.8%	-	-	-	1.8%	11.8%	7.39	0.003	1.94–28.1
BPD	8.8%	35.3%	5.64	0.002	1.89–16.8	9.5%	22.4%	2.74	0.008	1.3–5.77
Depressive	55.9%	38.2%	0.49	0.095	0.21–1.13	70.8%	60.5%	0.63	0.11	0.36–1.11
OPD	19.1%	17.6%	0.91	0.86	0.31–2.64	20.8%	21.1%	1.01	0.97	0.52–1.97
PSD	1.5%	5.9%	-	-	-	1.8%	6.6%	3.97	0.069	0.90–16.6
PTSD	32.4%	26.5%	0.75	0.54	0.30–1.88	17.9%	13.2%	0.70	0.36	0.32–1.51
Substance Use	5.9%	5.9%	1.00	1.00	0.17–5.75	15.5%	9.2%	0.55	0.19	0.23–1.34
**Psychotherapy**	**No**	**Yes**	**OR**	***p***	**95% CI**	**No**	**Yes**	**OR**	***p***	**95% CI**
**Disorder**										
Eating	0.0%	1.5%	-	-	-	0.0%	3.1%	-	-	-
Anxiety	7.7%	7.7%	1.00	1.00	0.43–4.44	3.3%	4.4%	1.34	0.68	0.34–5.30
Bipolar	0.0%	4.6%	-	-	-	3.3%	5.7%	1.74	0.42	0.46–6.60
BPD	7.7%	23.1%	3.60	0.056	0.97–13.4	4.4%	18.9%	5.00	0.003	1.7–14.7
Depressive	51.3%	50.8%	0.98	0.96	0.44–2.17	60.0%	70.4%	1.59	0.09	0.92–2.73
OPD	7.7%	24.6%	3.92	0.040	1.06–14.5	16.7%	23.9%	1.57	0.18	0.81–3.05
PSD	2.6%	3.1%	-	-	-	2.2%	3.1%	1.43	0.67	0.27–7.52
PTSD	38.5%	24.6%	0.52	0.14	0.22–1.23	25.6%	1.3%	0.37	0.004	0.19–0.74
Substance Use	10.3%	4.6%	0.42	0.28	0.09–2.00	20.0%	9.4%	0.42	0.021	0.20–0.87

BPD = Borderline Personality Disorder, OPD = Other than borderline Personality Disorder, PSD = Psychotic Spectrum Disorder, PTSD = Post-Traumatic Stress Disorder.

**Table 4 ijerph-17-08733-t004:** Frequencies of disorder per previous psychiatric hospitalization (PPH) and odds ratio per age group (-: no statistical significance).

	Youth					Adults				
PPH	No	Yes	OR	*p*	95%CI	No	Yes	OR	*p*	95%CI
**Disorder**										
Anxiety	5.5%	12.5%	2.46	0.224	0.58–10.55	4.4%	3.3%	0.74	0.664	0.19–2.92
Bipolar	2.7%	3.1%	-	-	-	4.4%	5.4%	1.26	0.704	0.39–4.08
BPD	11.0%	31.3%	3.69	0.015	1.3–10.5	5.6%	27.2%	6.26	0.000	2.77–14.14
Depressive	52.1%	50.0%	0.92	0.846	0.4–2.12	67.5%	66.3%	0.95	0.846	0.55–1.63
OPD	21.9%	9.4%	0.37	0.136	0.1–1.37	18.8%	25.0%	1.44	0.243	0.78–2.68
PSD	2.7%	3.1%	-	-	-	3.1%	3.3%	1.04	0.953	0.24–4.48
PTSD	34.2%	18.8%	0.44	0.114	0.16–1.22	21.3%	7.6%	0.31	0.007	0.13–0.72
Substance Use	6.8%	6.3%	0.91	0.910	0.17–4.94	10.0%	18.5%	2.04	0.058	0.98–4.27

BPD = Borderline Personality Disorder, OPD = Other than borderline Personality Disorder, PSD = Psychotic Spectrum Disorder, PTSD = Post-Traumatic Stress Disorder.

**Table 5 ijerph-17-08733-t005:** Frequencies of disorder per hospitalization reason, and adjustment Khi2 within age group (-: no statistical significance).

	Youth					Adults				
Reason	OAS	SI	SB	Khi2	*p*	OAS	SI	SB	Khi2	*p*
**Disorder**										
BPD	-	-	-	-	-	8.9%	14.0%	16.4%	1.31	0.52
Depressive	45.5%	47.2%	63.3%	0.73	0.69	69.6%	65.9%	67.2%	0.97	0.62
OPD	-	-	-	-	-	10.7%	27.1%	17.9%	3.87	0.14
PTSD	36.4%	32.1%	20.0	1.79	0.41	16.1%	19.4%	10.4%	1.47	0.48
Substance Use	-	-	-	-	-	8.9%	16.3%	10.4%	1.29	0.52

OAS = Other acute signs, SI = Suicidal Ideations, SB = Suicidal Behaviors, BPD = Borderline Personality Disorder, OPD = Other than borderline Personality Disorder, PTSD = Post-Traumatic Stress Disorder.

**Table 6 ijerph-17-08733-t006:** Frequencies of treatment per previous psychiatric hospitalization (PPH) and odds ratio per age group.

	Youth					Adults				
PPH	No	Yes	OR	*p*	95% CI	No	Yes	OR	*p*	95% CI
**Treatments**										
Antidepressants	28.6%	33.3%	1.25	0.634	0.5–3.13	37.7%	35.5%	0.908	0.720	0.54–1.54
Anxiolytics	15.6%	36.8%	3.14	0.037	1.07–9.19	33.3%	37.2%	1.18	0.646	0.57–2.44
Neuroleptics	20.6%	50.0%	3.86	0.003	1.58–9.42	31.5%	47.4%	1.95	0.018	1.12–3.4
Psychotherapy	12.8%	40.0%	4.53	0.005	1.57–13.1	20.0%	46.5%	3.48	0.000	1.9–6.37

**Table 7 ijerph-17-08733-t007:** Frequencies of treatment per hospitalization reason and adjustment Khi2 within age group.

	Youth						Adults					
Reason	OAS	SI	SB	Khi2	df	*p*	OAS	SI	SB	Khi2	df	*p*
**Treatment**												
Antidepressants	28.6%	26.5%	36.7%	0.94	2	0.63	68.5%	49.2%	60.9%	6.42	2	0.04
Anxiolytics	66.7%	67.3%	70.0%	0.08	2	0.96	87.3%	81.0%	87.7%	1.98	2	0.37
Neuroleptics	40.9%	42.0%	13.3%	7.66	2	0.02	30.9%	35.2%	23.4%	2.73	2	0.25
Psychotherapeutic	63.6%	57.7%	70.0%	1.24	2	0.54	70.9%	62.0%	61.5%	1.53	2	0.47

OAS = Other acute signs, SI = Suicidal Ideations, SB = Suicidal Behaviors.

**Table 8 ijerph-17-08733-t008:** Frequencies of reason for hospitalization for people with previous psychiatric hospitalization, OR between age groups, and adjustment Khi2 within age group.

Reason	Youth	Adults	OR	*p*	95% CI
Other acute signs	12.5%	20.7%	0.433	0.177	0.16–1.20
Ideation	50.0%	47.8%	0.835	0.610	0.74–1.49
Behavior	37.5%	31.5%	0.874	0.762	0.42–1.82
Khi2	7.00	10.3			
Df	2	2			
*p*	0.030	0.006

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
