# Peer review of "Clinical Characteristics of Suicidal Youths and Adults: A One-Year Retrospective Study"

_ijerph, 2020, doi:10.3390/ijerph17238733_

Round 1

Reviewer 1 Report

The current study sought to assess the specificities and similarities between suicidal youth and adults in order to ascertain whether there are needs for adaptation or opportunities for sharing methods and treatments between these two different stages of life. It is a very relevant topic in the clinical field and it is my opinion that the current study should be commended for broadening the insights on a very relevant clinical concern. I feel that the current study could be considered for publication but needs major revisions before it can be accepted for publication.

  • First of all, although I am a Doors fan myself, I do not like the use of references from pop culture in a scientific manuscript. Authors of the manuscript should remove lines 34-38 and all the other references to Morrison’s life used in the manuscript.

  • Authors of the present manuscript mostly used expressions such as “Our results suggest” (line 28) and “Our clinical and scientific interest” (line 42). My advice is to seek a more neutral language by avoiding using "he, she," or "he/she" and “we/our”. So for instance, the aforementioned examples could be rephrased “Results of the present study suggest” and “Clinical and scientific interest of this study”.

  • The authors stated that “Suicidality is frequently associated with mental disorders, such as anxiety, mood disorders, substance abuse, etc, as well as with adverse social and interpersonal conditions, making it a phenomena that is difficult to predict and to prevent” (lines 48-50) and I think this is a great point. Nevertheless, the authors did not deepen the potential predictors of suicidal behaviors. The authors should provide a brief overview of the antecedents highlighted by previous literature of suicidal behaviors. Specifically, the Authors should focus on Emotional Intelligence (both as ability and trait) as a robust line of studies suggested that higher levels of emotional intelligence (again both trait and ability) may decrease the likelihood of suicidal behavior. The following are important references that can be used to address this issue:
    - Domínguez-García, E., & Fernández-Berrocal, P. (2018). The Association Between Emotional Intelligence and Suicidal Behavior: A Systematic Review. Frontiers in Psychology, 9. doi:10.3389/fpsyg.2018.02380
  • Mikolajczak, M., Petrides, K. V., & Hurry, J. (2009). Adolescents choosing self-harm as an emotion regulation strategy: The protective role of trait emotional intelligence. British Journal of Clinical Psychology, 48(2), 181–193. doi:10.1348/014466508x386027
  • Barberis, N., Verrastro, V., Papa, F., & Quattropani, M. (2020). Suicidal ideation and psychological control in emerging adults: The role of trait EI. MALTRATTAMENTO E ABUSO ALL’INFANZIA, (2), 13–28. doi:10.3280/mal2020-002002

  • The authors stated that “young people would be characterized by less frequent personality disorders, less pharmacological treatments, and less relapses, but a higher number of hospitalizations due to suicidal behaviors in compare with ideations or states of extreme vulnerability” (lines 81-82). Although I am partially sympathetic with such hypothesis, authors should deepen and elaborate in the introduction section why such results were expected, specifically by using proper references. For instance, why do the authors expect fewer drug treatments and fewer relapses, but more hospitalizations in younger subjects? I think that the authors need to put together a clearer rationale first, present it, then walk the reader through the observed dimensions, in accordance with the aforementioned aims and hypothesis.

  • The authors stated that “As health services are often organized separately for adults and for children, adolescents and young adults in suicidal crisis can often be admitted to adult services. In this study, we aimed to identify the similarities and specificities of a group of young people and a group of mature adults” (lines 74-76) and “we have considered two age groups: patients aged 16 to 26 years, 110 whom we called “youths”, and patients aged 27 years and above, whom we called “adults” (lines 110-111) What is the definition of "young adult" that the authors used? Then, were they specifically interested in using a certain developmental perspective? Please elaborate.

  • The authors stated that ”In terms of treatments, opposite to our expectations, we observed similar frequencies of neuroleptic treatments in youths as in adults” (line 334-335). I think that authors should really elaborate on this divergent result deeply.

  • A very serious limitation that I find in this study, and which I think should be mentioned together with the others, is the high degree of presence of patients with more than one diagnosis (youth 39% had more than one disorder and 48.4% of adults; line 194). Although the authors have provided an explanation for the influence of all nine diagnostic categories listed, how is it possible to explain the influence of different diseases in patients with more than one diagnosis? The authors should include the high presence of patients who had more than one disorder as another limitation of the study, as it may potentially confound the explanation of the underlying dysfunctional functioning in individuals who attempted suicide.

In summary, I hope that the authors do not feel intimidated by my recommendations for major revisions. I am convinced that the authors have an interesting study with a lot of potential. I am confident that with a little bit of work and clarification this paper may be strengthened. I am looking forward to reviewing the edited version of the paper.

Author Response

The current study sought to assess the specificities and similarities between suicidal youth and adults in order to ascertain whether there are needs for adaptation or opportunities for sharing methods and treatments between these two different stages of life. It is a very relevant topic in the clinical field and it is my opinion that the current study should be commended for broadening the insights on a very relevant clinical concern. I feel that the current study could be considered for publication but needs major revisions before it can be accepted for publication.

• First of all, although I am a Doors fan myself, I do not like the use of references from pop culture in a scientific manuscript. Authors of the manuscript should remove lines 34-38 and all the other references to Morrison’s life used in the manuscript.

We employ this introduction to introduce a subject that is both the object of science and universal in terms of human experience, which is that suffering psychologically during youth may lead to very dark and tragic fates. The reference to the Doors achieves this efficiently. Therefore, we are rather attached to starting the manuscript in this way. But we can edit out if the reviewers feel strongly that this impedes the quality of the manuscript, which we feel is much improved after considering all the comments of the reviewers

• Authors of the present manuscript mostly used expressions such as “Our results suggest” (line 28) and “Our clinical and scientific interest” (line 42). My advice is to seek a more neutral language by avoiding using "he, she," or "he/she" and “we/our”. So for instance, the aforementioned examples could be rephrased “Results of the present study suggest” and “Clinical and scientific interest of this study”.

OK, we correct lines 28 and 42 as you mentioned,

• The authors stated that “Suicidality is frequently associated with mental disorders, such as anxiety, mood disorders, substance abuse, etc, as well as with adverse social and interpersonal conditions, making it a phenomena that is difficult to predict and to prevent” (lines 48-50) and I think this is a great point. Nevertheless, the authors did not deepen the potential predictors of suicidal behaviors. The authors should provide a brief overview of the antecedents highlighted by previous literature of suicidal behaviors. Specifically, the Authors should focus on Emotional Intelligence (both as ability and trait) as a robust line of studies suggested that higher levels of emotional intelligence (again both trait and ability) may decrease the likelihood of suicidal behavior. The following are important references that can be used to address this issue: - Domínguez-García, E., & Fernández-Berrocal, P. (2018). The Association Between Emotional Intelligence and Suicidal Behavior: A Systematic Review. Frontiers in Psychology, 9. doi:10.3389/fpsyg.2018.02380 - Mikolajczak, M., Petrides, K. V., & Hurry, J. (2009). Adolescents choosing self-harm as an emotion regulation strategy: The protective role of trait emotional intelligence. British Journal of Clinical Psychology, 48(2), 181–193. doi:10.1348/014466508x386027 - Barberis, N., Verrastro, V., Papa, F., & Quattropani, M. (2020). Suicidal ideation and psychological control in emerging adults: The role of trait EI. MALTRATTAMENTO E ABUSO ALL’INFANZIA, (2), 13–28. doi:10.3280/mal2020-002002

Thank you for your comment which indeed brings up the very relevant point of protective factors. We therefore:
1. Line 54: Complete our introduction to mention the protective role of emotional intelligence and emotional regulation skills (with reference to Domínguez-García & Fernández-Berrocal and Mikolajczak and al., Bateman 1999)
2. Line 421 of the new version of the manuscript: Complete our discussion to suggest psychotherapeutic intervention targeting emotion regulation.

The authors stated that “young people would be characterized by less frequent personality disorders, less pharmacological treatments, less relapses, higher number of hospitalizations due to suicidal behaviors in compare with ideations or states of extreme vulnerability” (lines 81-82). Although I am partially sympathetic with such hypothesis, authors should deepen and elaborate in the introduction section why such results were expected, specifically by using proper references. For instance, why do the authors expect fewer drug treatments and fewer relapses, but more hospitalizations in younger subjects? I think that the authors need to put together a clearer rationale first, present it, then walk the reader through the observed dimensions, in accordance with the aforementioned aims and hypothesis.

Thank you for this thoughtful comment which brought back to mind our original observation, that direct comparisons of psychiatric diagnoses of suicidal youths and adults are seldom examined in the literature. We agree that we need to make our rationale clearer, by stating more direct reference to lifespan research of developmental psychopathology which can be linked to more categorical disorders such as mood, psychotic, personality or substance use disorders in adulthood (Sroufe & Rutter 1984, Kessler 2005, Jones 2013). Importantly, our hypotheses stem from evidence of psychiatric diagnosing practices, and the extent of psychopharmacological prescriptions as described in the literature. We formulated our hypotheses on young people as follows:
- less personality disorders (PD): even though some studies state that PDs are as frequent in youth as in adults (Grilo 1998, Stevenson 2010, Chanen 2013), most studies simply do not report on PDs (Alonso 2004, Brent 2006, Kessler 2011). In this context, we also consider the documented reluctance to make a diagnosis of PD, for a young patient (Biskin 2013). Therefore, we made the hypothesis that PD would have been less frequently diagnosed in our young population,
- less pharmacological treatments: our expectations are based on relevant studies, for example Turecki (2016) who warns of the risks of using antidepressants and neuroleptics with young people, and Brent (2016) who expresses similar warnings, and formulates that a wide range of other forms of treatments should be used by clinicians dealing with suicidal youths.
- as frequent relapses as for adults: despite later onset of disorders associated with suicidality and high relapse rate (Bertolote 2004, Chesney 2014, Besch 2020), recent studies indicate high relapse rate in youths (Hawton 2012, Cha 2018),
- higher number of reasons for current hospitalization due to suicidal behaviors in comparison to suicidal ideations: youth being characterised by more impulsivity and conduct disorders (Brent 1999, Kessler 2005) i.e. externalised symptoms, we expected to observe an uneven distribution in “reasons for hospitalization”, in the direction of more current hospitalization for suicidal behaviors in comparison to suicidal ideations, whereas in adults we anticipated a more balanced distribution (also Cripps 2020).

From first version lines 81-85 : “We expected that young people would be characterized by less frequent personality disorders, less pharmacological treatments, and less relapses, but a higher number of hospitalizations due to suicidal behaviors in compare with ideations or states of extreme vulnerability. Further, we wished to explore specific age-related associations between variables related to suicidality”.

…we now deepened and referenced our hypotheses in version 2, lines 97 to 128

“This study is theoretically and empirically framed on Sroufe & Rutter’s principles of developmental psychopathology [21], on the results of Kessler [1] and Jones [22] regarding mental disorders’ age of onset and development through life course, as well as on McGorry’s contemporary perspective of needs for youth mental health [18]. Sroufe & Rutter’s principles of developmental psychopathology [21] are especially relevant when considering age-related differences in mental disorders. For instance, results from life course studies indicate that anxiety and conduct disorders usually appear earlier in childhood [22, 23], whilst mood, substance use, and personality disorders appear later during adolescence and further develop between the 20s and 40s [1, 24, 25]. Across the lifespan, suicidal behaviour may represent a maladaptive coping strategy more likely activated in stressful situations. This illustrates both homotypic and heterotypic continuity in a developmental psychopathology framework.
As regards personality disorders, although studies indicate that they can be reliably assessed in young people [25, 26], that they represent an important risk factor for suicidality [27, 28], most studies examining the repartition of mental disorders with age do not include diagnoses of PD [1, 19, 29, 30]. Indeed, research documents the reluctance of clinicians to give youths PD [23, 31, 32]. From the perspective of developmental studies that do report on PDs, it appears that their prevalence increases from youth to adulthood [24].
Based on epidemiological and developmental psychology studies, we therefore expect that PDs should be less frequently diagnosed in youths in comparison to adults presenting at emergency psychiatric services. Furthermore, we expect that anxiety disorder should be more frequent in youth, whereas substance use disorders less frequent. Finally, we expect that frequency of mood disorders should be equivalent in youth and adult presenting at the psychiatric emergency service [29, 30, 33].
In terms of treatment, it is expected that youths would present at psychiatric emergency in similar ratios of ongoing psychotherapeutic treatment in comparison to adults [34], and generally, with less ongoing pharmacological treatments [35, 36], in particular less neuroleptics [37, 38], and less antidepressants due to warnings on their iatrogenic effect in suicidal youths [39]. Concerning previous hospitalizations, despite the logical fallacy that the older you are the more time you had to relapse, research indicates that suicidal relapse is as frequent among young people as it is among adults [40, 41]. Finally, concerning reasons of hospitalization, higher prevalence of impulsivity and conduct disorders in youths support the hypothesis that they would present at the psychiatric emergency more frequently for suicidal behaviors rather than suicidal ideations, while adults would have a more balanced distribution between behaviors and ideations [42].”

The authors stated that “As health services are often organized separately for adults and for children, adolescents and young adults in suicidal crisis can often be admitted to adult services. In this study, we aimed to identify the similarities and specificities of a group of young people and a group of mature adults” (lines 74-76) and “we have considered two age groups: patients aged 16 to 26 years, 110 whom we called “youths”, and patients aged 27 years and above, whom we called “adults” (lines 110-111).
What is the definition of "young adult" that the authors used ? Then, were they specifically interested in using a certain developmental perspective? Please elaborate.

In the perspective of McGorry, we are interested in young people, or youths, aged from the end of childhood until mid-twenties, i.e. 12 to 25. This period overlaps adolescence and young adulthood, and is crucial for mental health but exposed to a break in the continuity of care which is classically organised whether for children and adolescents or for adults, with a threshold at 18. According to Fusar-Poli (2019), “Young adult” correspond the subgroup of youths aged 19 to 25, and we prefer to get rid of this term which may bring confusion.
The manuscript is amended accordingly at line 70-71.

• The authors stated that ”In terms of treatments, opposite to our expectations, we observed similar frequencies of neuroleptic treatments in youths as in adults” (line 334-335). I think that authors should really elaborate on this divergent result deeply.

This result is questioning and requires interpretation. Lower rate of antidepressants is likely attributable to the mentioned warnings on their possible iatrogenic effect in youths. Neuroleptics rate found higher than expected is consistent with their use to address conduct disorders and disruptive behaviors as well as depression in youths (Aderhold 2015). Therefore, the fact that young people treated with neuroleptics are less often hospitalised for suicidal behaviour than for ideas or warning signs might be considered positive as it suggests some success in preventing the transition to an actual suicide attempt which would increase the risk of relapse and of a fatal issue. Yet, youths are still the most hospitalized for suicidal risk, which questions on the recognition of their depressive symptomatology, unexpectedly lower than adults, and their high relapse rate. Considering the difficulties to treat suicidality in youth with pharmacological means (Katz 2020), one possible transdiagnostic component of suicidality would be emotion regulation, as a core component cutting across the multiple clinical correlates of suicidality: anhedonia and depression (Cha 2018), emotions intensity and lability in BPD (Bateman 1999), deficit in emotional intelligence (Domínguez-García & Fernández-Berrocal 2018, Mikolajczak and al 2009)
The manuscript is amended accordingly at line 418-420.

• A very serious limitation that I find in this study, and which I think should be mentioned together with the others, is the high degree of presence of patients with more than one diagnosis (youth 39% had more than one disorder and 48.4% of adults; line 194). Although the authors have provided an explanation for the influence of all nine diagnostic categories listed, how is it possible to explain the influence of different diseases in patients with more than one diagnosis? The authors should include the high presence of patients who had more than one disorder as another limitation of the study, as it may potentially confound the explanation of the underlying dysfunctional functioning in individuals who attempted suicide.

As comorbidity is common in the daily experience with our cohort population, we thought necessary to mention it as a descriptive fact. However, the question "how is it possible to explain the influence of different diseases in patients with more than one diagnosis?" is fully relevant. Answers could be sought by analysing how combination of disorders are associated with the other variables (treatments, previous hospitalisation, reasons of current hospitalization). Another way could be factorial analysis, similar to studies suggesting the existence of a latent general factor of psychopathology that might explain comorbidity. However, this would represent a different approach with a substantial amount of additional results, therefore we sure keep in mind your request, but find it preferable to work on this point for future publication.
Following your comment, we mention comorbidity as a limit of this study, and open to the relevance to investigate comorbidity in the context of suicidality with factorial analysis
The manuscript is amended accordingly at line 450-454.

In summary, I hope that the authors do not feel intimidated by my recommendations for major revisions. I am convinced that the authors have an interesting study with a lot of potential. I am confident that with a little bit of work and clarification this paper may be strengthened. I am looking forward to reviewing the edited version of the paper.

No problem, we warmly thank you for your comments that are of great help to strengthen the theoretical frame of this study and make its narrative more clear.

References:
• Sroufe, A.; Rutter, M. The Domain of Developmental Psychopathology. Child Development 1984 55(1) pp. 17-29
• Jones, P.B. Adult mental health disorders and their age at onset. The British Journal of Psychiatry 2013 202, s5-s10.
• Grilo, C. M.; McGlashan, T. H.; Quinlan, D. M.; Walker, M. L.; Greenfeld,D. G.; Edell, W. S. Frequency of Personality Disorders in Two Age Cohorts of Psychiatric Inpatients Am J Psychiatry. 1998 155:140–142
• Fusar-Poli, P. Integrated Mental Health Services for the Developmental Period (0 to 25 Years): A Critical Review of the Evidence. Frontiers in Psychiatry 2019 10(355).
• Chanen, A. M.; McCutcheon, L. Prevention and early intervention for borderline personality disorder: current status and recent evidence. The British Journal of Psychiatry 2013 202, s24–s29
• Stevenson, J.; Datyner, A.; Boyce, P.; Brodaty, H. The effect of age on prevalence, type and diagnosis of personality disorder in psychiatric inpatients. Int J Geriatr Psychiatry. 2011 26 981–987.
• Alonso, J.; Angermeyer, M.C.; Bernert, S.; Bruffaerts, R.; Brugha, T.S.; Bryson, H.; de Girolamo G.; de Graaf, R.; Demyttenaere, K.; Gasquet, I.; Haro, J.M.; Katz, S.J.; Kessler, R.C.; Kovess, V.; Lépine, J.P.; Ormel, J.; Polidori, G.; Russo, L.J.; Vilagut, G.; Almansa, J.; Arbabzadeh‐Bouchez, S.; Autonell, J.; Bernal, M.; Buist‐Bouwman, M.A.; Codony, M.; Domingo‐Salvany, A.; Ferrer, M.; Joo, S.S.; Martínez‐Alonso, M.; Matschinger, H.; Mazzi, F.; Morgan, Z.; Morosini, P.; Palacín, C.; Romera, B.; Taub, N.; Vollebergh, W.A.M. Prevalence of mental disorders in Europe: results from the European Study of the Epidemiology of Mental Disorders (ESEMeD) project. Acta Psychiatrica Scandinavica. 2004 109: 21-27
• Bridge, J.A.; Goldstein, T. R.; Brent, D. A. Adolescent suicide and suicidal behavior. Journal of Child Psychology and Psychiatry. 2006 47:3/4 pp 372–394
• Biskin, R.S. Treatment of Borderline Personality Disorder in Youth. J Can Acad Child Adolesc Psychiatry. 2013 22(3) 230-234
• Gore, F. M.; Bloem, P. J. N.; Patton, G. C.; Ferguson, J.; Joseph, V.; Coffey, C.; Sawyer, S. M.; Mathers, C. D. Global burden of disease in young people aged 10–24 years: a systematic analysis. Lancet 2011 377, 2093–102.

• Besch, V.; Debbané, M. ; Greiner, C.; Magnin, C.; De Néris, M.; Ambrosetti, J.; Perroud, N.; Poulet, E. ; Prada, P. Emergency psychiatric management of borderline personality disorder: Towards an articulation of modalities for personalised integrative care. Encéphale 2020.
• Hawton, K.; Bergen, H.; Kapur, N.; Cooper, J.; Steeg, S.; Ness, J.; Waters, K. Repetition of self-harm and suicide following self-harm in children and adolescents: findings from the Multicentre Study of Self-harm in England. Journal of Child Psychology and Psychiatry. 2012 53(12) pp 1212–1219
• Cha, C. B.; Franz, P. J.; Guzman, E. M.; Glenn, C. R.; Kleiman, E. M.; Nock, M. K. Annual Research Review: Suicide among youth – epidemiology, (potential) etiology, and treatment. Journal of Child Psychology and Psychiatry. 2018 59(4) pp 460–482
• Olfson, M.; Blanco, C.; Liu, S. M. ; Wang, S.; Correll, C. U. National Trends in the Office-Based Treatment of Children, Adolescents, and Adults With Antipsychotics. Arch Gen Psychiatry. 2012 69(12) 1247-1256.
• Katz, C.; Randall, J. R.; Leong, C.; Sareen, J.; Bolton, J. M. Psychotropic medication use before and after suicidal presentations to the emergency department: A longitudinal analysis. General Hospital Psychiatry. 2020 63, 68–75.
• Greenfield, B.; Henry, M.; Lis, E.; Slatkoff, J.; Guile, JM.; Dougherty, G.; Zhang, X.; Raz, A.; Eugene Arnold, L.; Daniel, L.; Mishara, B. L.; Koenekoop, R., K.; de CastroCorrelates, F. Stability and predictors of borderline personality disorder among previously suicidal youth. Eur Child Adolesc Psychiatry. 2015 24, 397–406

Reviewer 2 Report

Besch and colleagues sought to identify the specificities and similarities between suicidal youths and adults to find out whether there are needs for adaptation or opportunities for sharing methods and treatments. Based on their findings, they suggest that young people could benefit from brief interventions conducted for older adults. This article touches on an important topic and may be of interest overall. However, the authors should address the following issues:

  1. lines 19-20, I can't understand the meaning of the sentence "...in order to ascertain whether there are needs for adaptation or opportunities for sharing methods and treatments
  2. lines 34-45, I wonder if talking about Jim Morrison is relevant in an article purporting to be scientific and if it wouldn’t be better to get straight to the point. 
  3. line 50, phenomena --> phenomenon 
  4. line 51, visible part --> consequence
  5. line 83, compare --> comparison
  6. lines 120-121, There is no table 2. It is misnamed table 4.
  7. lines 136-137, It is not clear to me what a warning sign is without explicit behaviors or ideations. Can you give some examples??
  8. lines 139-141, The meaning of this sentence "The benefit is to transform the clinical reasons for which patients were hospitalized into a standardized variable which can be used in the frame of theoretical models (table 1)." is obscure to me. The authors should clarify it. 
  9. line 144, What do you mean for the “diffuse” risk factor?
  10. line 146, What examples are the authors referring to?
  11. lines 149-150, Please clarify the sentence “Rather than considering some of these variables as inputs and other as outcomes in a merely linear causal mechanism“
  12. line 194, Do you mean that "39% of young people and 48.4% of adults had more than one disorder"?
  13. In the discussion, the authors often say that the results are contrary to their expectations, but then they do not clarify either on what their expectations are founded or the possible theoretical and practical implications of their violations. So what?
  14. lines 404-406, The authors should clarify which “adult care system” they refer to. A work like the present cannot represent evidence of the appropriateness/inappropriateness of any care system since it is not a clinical trial. Thus, the sentence "...from a statistical point of view and in our context, there is no evidence that an adult care system would be inappropriate for the care of young people", while formally correct, seems empty of actual information content.

Reviewer 3 Report

line 46: It would be definitely useful/ illustrative to state also which is the first one.

Line 51: I suggest here the use of the singular form for the word "behaviour"

Line 52: Please, start a new paragraph for the explanation of the phases/ stages

Line 62: use letters for the number "2"

Line 62: A footnote explaining what "Gaussian" means would be helpful for the reader.

Lines 61- 55: I suggest rephrasing these sentences. Meaning is quite obscure.

Lines 67-69: I consider here a theoretical support to be necessary  to support  this affirmation.

Line 71: Place a comma after "care"

Line 74: Place a comma after "trajectories"

Line 74: Please, start a new paragraph from "As health..." 

Line 75: The plural form for "crisis" is "crises"

Lines 78- 87: OK! You have one major objective and several minor objectives, a starting premise, but why are you caring out this study? Which are your major aims? What for are you studying this? What do you want to prove? It is very important to state this clearly in this first stage of your research. Also, make sure you comment on this in your conclusion section.

Line 199: I suggest capitalising "Distribution" and also mentioning here clearly the age range for the study

Line 320: I was expecting, at some moment, a possible explanation for the fact that women had a higher prevalence of hospitalisation. Make sure you do not only present your results, but you also interpret them. I perceive this as being a minus for your chapter, since at times only numbers are presented and no possible interpretation is offered to establish connections between statistics and the causes standing behind results.

Line 341: You claim this several times all along the article. Why? You should explain it. You should be also careful with this possible logical fallacy. Without clear theoretical support in this direction, your article can be mined.

Line 379: Place a comma after "(table 1)"

Line 402: Avoid when possible, the use of "we". Substitute this pronoun with  "this study/ paper/research"

Line 402: Leave out the word "tries", since it rests this study trustfulness. You seem to question your own paper and the veracity of this research.  Also, instead of "give" use "provide".

Reviewer 4 Report

1.

In our study, we have considered all patients admitted, without restriction, for a total of 357 106patientsaged 16 to 74years, out of which 254 were women, 31 were adolescents according to WHO 107definition (less than 20 years old, i.e. teen-agers) and 74 were aged between 20 and 26years. In across-108sectionalapproach, and following literature on critical period for the onset of mental disorders, and 109on the age of first suicide attempts, we have considered two age groups: patients aged 16 to 26 years, 110whom we called “youths”, and patients aged 27 years and above, whom we called “adults”.

Comment:

In my opinion, the division of respondents into adolescents and adults is unjustified and wrong. The age of 16 to 20 is the period of adolescence. At that time, mental development is specific and some dimensions, e.g. personality, are only just being shaped (up to around 25 years of age). In my opinion, treating the study group 16-26 as homogeneous is a serious methodological error. Since the research assumptions made in the development perspective, they should have been guided by the findings of developmental psychology.

the introduction should justify such a selection of respondents to the following groups: adolescents (16-26) and adults (powuże 27)     This calls into question the scientific value of the research carried out.

Author Response

In our study, we have considered all patients admitted, without restriction, for a total of 357 patients aged 16 to 74 years, out of which 254 were women, 31 were adolescents according to WHO definition (less than 20 years old, i.e. teen-agers) and 74 were aged between 20 and 26 years. In a cross-sectional approach, and following literature on critical period for the onset of mental disorders, and on the age of first suicide attempts, we have considered two age groups: patients aged 16 to 26 years, whom we called “youths”, and patients aged 27 years and above, whom we called “adults”.
Comment:
In my opinion, the division of respondents into adolescents and adults is unjustified and wrong. The age of 16 to 20 is the period of adolescence. At that time, mental development is specific and some dimensions, e.g. personality, are only just being shaped (up to around 25 years of age). In my opinion, treating the study group 16-26 as homogeneous is a serious methodological error.
Since the research assumptions made in the development perspective, they should have been guided by the findings of developmental psychology. The introduction should justify such a selection of respondents to the following groups: adolescents (16-26) and adults (powuże 27). This calls into question the scientific value of the research carried out.

Thank you for this comment, which shows the need to clarify the object of this study. Following McGorry and others, it is because things are being “shaped” until approximately 25 that this group can legitimately be put together. We therefore refer to this period as “youth”. See further references below for theoretical and conceptual justification to our division of respondents.
Concerning brain development and mental health, literature converges to delimitate “youth” between 12 and 25 (see Fusar-Poli 2019 for a review). Concerning suicidality, several studies show that it peaks at the age of 20, dips at 26, and rebounds at 38 (Slama 2009, Cash 2018, O’Hare forthcoming). That’s why we separated our sample population into one group of patients aged 26 and less, whom we call "youth", and one group aged over 26 years whom we call "adults". Doing this, our “youth” category deliberately overlaps on the usual groupings into “adolescents” and “adults” that match with traditional definition and care organisations, and it is our intent to do so in order to contribute to “strengthening the weakest link in the public mental health system” (McGorry 2007, Jones 2013).
Following your recommendation, we explain reasons for this selection in the introduction at lines 70-84, and replace the term “adolescence” in the text everywhere as it fosters confusion. Many thanks for your straightforward comments.

• Fusar-Poli, P. Integrated Mental Health Services for the Developmental Period (0 to 25 Years): A Critical Review of the Evidence. Frontiers in Psychiatry 2019 10(355).
• Cash, S. J.; Bridge, J. A. Epidemiology of Youth Suicide and Suicidal Behavior. Curr Opin Pediatr. 2009 21(5) 613–619
• Jones, P.B. Adult mental health disorders and their age at onset. The British Journal of Psychiatry 2013 202, s5-s10.

Round 2

Reviewer 1 Report

I have read the previous version of this manuscript that sought to assess the specificities and similarities between suicidal youth and adults in order to ascertain whether there are needs for adaptation or opportunities for sharing methods and treatments between these two different stages of life. I have some more confidence in the paper now, as it provides a clearer rationale and a stronger theoretical framework. Thanks for making all the revisions. I feel that the current version of the manuscript could be accepted for publication.